# Inferring *bona fide* Differentially Expressed Genes and Their Variants Associated with Vitamin K Deficiency Using a Systems Genetics Approach

**DOI:** 10.3390/genes13112078

**Published:** 2022-11-09

**Authors:** Shalini Rajagopal, Akanksha Sharma, Anita Simlot, Praveen Mathur, Sudhir Mehta, Sumita Mehta, Jalaja Naravula, Krishna Mohan Medicherla, Anil Kumar, Uma Kanga, Renuka Suravajhala, Ramji Kumar Bhandari, Bipin G. Nair, P. B. Kavi Kishor, Prashanth Suravajhala

**Affiliations:** 1Amrita School of Biotechnology, Amrita Vishwa Vidyapeetham, Clappana 690525, India; 2Department of Gynecology and Obstetrics, Babu Jagjivan Ram Memorial Hospital, Delhi 110033, India; 3Bioclues.org, Hyderabad 500072, India; 4Department of Transplant Immunology and Immunogenetics, AIIMS, New Delhi 110029, India; 5Department of Gynaecology and Obstetrics, Pandit Deen Dayal Upadhyaya Hospital (Gangori), Jaipur 302007, India; 6Department of Pediatric Surgery, SMS Medical College, JLN Marg, Jaipur 302004, India; 7Department of Medicine, SMS Medical College, JLN Marg, Jaipur 302004, India; 8Department of Biotechnology, Vignan’s Foundation for Science, Technology & Research, Vadlamudi, Guntur 522213, India; 9Department of Biotechnology and Bioinformatics, Birla Institute of Scientific Research, Statue Circle, Jaipur 302001, India; 10Department of Biotechnology Birla Institute of Technology Mesra, Off Campus Jaipur, Jaipur 302017, India; 11Department of Biology, University of North Carolina Greensboro, Greensboro, NC 27412, USA

**Keywords:** RNA-Seq, vitamin K, comorbidities, differentially expressed genes, variant analysis

## Abstract

Systems genetics is key for integrating a large number of variants associated with diseases. Vitamin K (VK) is one of the scarcely studied disease conditions. In this work, we ascertained the differentially expressed genes (DEGs) and variants associated with individual subpopulations of VK disease phenotypes, *viz*., myocardial infarction, renal failure and prostate cancer. We sought to ask whether or not any DEGs harbor pathogenic variants common in these conditions, attempt to bridge the gap in finding characteristic biomarkers and discuss the role of long noncoding RNAs (lncRNAs) in the biogenesis of VK deficiencies.

## 1. Introduction

The next-generation sequencing (NGS) technologies paved the way for systems genomics [1]. As NGS has provided a scope for understanding novel biological mechanisms and the molecular underpinning of complex diseases, thorough genomic and transcriptome analyses are needed [2]. With RNA-Seq being used for investigating the dynamic nature of the cell’s transcriptome, which is the component of the genome that is actively translated into RNA molecules, researchers are able to predict when and where genes are turned on or off in a range of cell types/situations. As the number of biological samples investigated using RNA-Seq analysis expands, the community has developed a wide range of bioinformatics tools to meet specific demands with highly optimized pipelines for further downstream processing [3]. There are a multitude of advantages of analyzing transcriptome data; for example, finding genomic features such as gene and transcript expression, miRNAs and non-coding RNAs (long noncoding RNAs, and small RNAs), in addition to predicting variants or mutations in the form of novel isoforms and SNPs (SNVs or indels) with sufficiently high expression levels.

In the recent past, blood disorders have been well studied using RNA-Seq, as, for example, [4] studied the potential gene expression difference in acute respiratory distress syndrome (ARDS) using hematopoietic stem cell transplantation, which implies differences in immune response and interferon signaling pathways. Zheng et al. recently provided a genome-wide analysis comparing the RNA-Seq data from hemorrhoidal diseases [5]. Of late, single-cell transcriptomic strategies have just begun to be understood and are being used to study various disease phenotypes, such as in [6,7], to mention a few. However, vitamin K (VK) deficiency is not studied extensively, and thus, the genome-wide transcriptome profiles are not known. There are, indeed, associated VK phenotypes such as thrombosis, thrombocytopenia, myocardial infarction, renal failure and prostate cancer, which are used to ascertain the differentially expressed gene (DEG) profiles [8]. Furthermore, calling the somatic or germline variants after the DEG profiles are checked could be a holistic measure for screening and characterizing biomarkers. Our study attempts to fill this gap wherein we have used myocardial, prostate cancer (both from the Sequence Read Archive) and renal (in-house) datasets and checked for the DEG profiles and candidate mutations associated with VK deficiency. We discuss the impending effects of the role of DEG profiles in the context of VK and associated deficiencies.

## 2. Materials and Methods

### 2.1. Datasets

The myocardial data comprise the strand-specific RNA-Seq dataset for both coding and noncoding RNA profiling from 28 hypertrophic cardiomyopathy (HCM) patients and 9 healthy donors [9]. For our analysis, we considered three control and three treatment samples from this study, obtained through the Sequence Read Archive (SRA: ncbi.nlm.nih.gov/sra, accessed on 9 June 2022). The following myocardial samples were taken: SRR8586402; SRR8586407; SRR8586429 (control) and SRR8586409; SRR8586423; SRR8586431 (treated). In addition, we used data from the complete transcriptome landscape of prostate cancer (PCa) using RNA-Seq from another study [10]: ERR031017; ERR031029; ERR031031 (control) and ERR031018; ERR031030; ERR031032 (treated), which we also compared with our own PCa datasets from our lab (PRJNA616165). Finally, the renal datasets were divided into three groups: rejection time point, well-functioning rejection matched, and post-therapy (PRJNA854340). There were four patients in the test group (R2, R4, R5 and R6), all of whom received post-therapy (R2_pt, R4_pt, R5_pt and R6_pt), and three patients in the control group who were in good health; without renal failure is abbreviated as WF (WF2, WF4 and WF6). All samples were analyzed using paired-end datasets, supplementing three of each pair of datasets (Appendix A).

### 2.2. RNA Sequencing Analysis, Statistics and Validation

The reads were checked for quality using FastQC [11], followed by HISAT2 [12,13], which was used to align them to the human genome (GRCh38 assembly). A Cufflinks-Cuffdiff pipeline was employed to find significant changes at the level of transcript expression, splicing and promoters [14], which was later benchmarked in our lab and used to run through the workflow [15]. As Cufflinks treats each pair of fragment reads as a single alignment, there is always an optimal amount of time and energy saved. With an “overlap graph”, each of the largest sets of reads originating from the same isoform results in a minimal set of fragments, and this determines the transcript abundances using a statistical model [16] (Figure 1).

RNA-Seq reads were trimmed, but there was no significant reduction in size; further aligned reads were processed to generate SAM, BAM and sorted BAM files through a cohort of tools. As the DEG analysis primarily relies on paired samples, we checked paired-end reads, i.e., control vs. treated in case of the myocardial, prostate and renal datasets. The resulting tables were filtered (after ensuring the Cufflinks pipeline was used with the -g option to check for novel isoforms) by *p*- and q-values of ≤0.05 and ≥2 log2-fold-change ≤2, respectively. The output BAM files were subjected to the consensus mapping of SNPs with different tools such as SAMtools [17,18], VarScan [19], FreeBayes [20], Vt [21] and GATK [22], using the default parameters such as the number of criteria, including coverage, read counts, *p*-value, variant allele frequency, base quality and the number of strands on which the variant was observed [23]. The filtered variants were then compared with the ClinVar database. FreeBayes is a haplotype-based and Bayesian genetic variant detector which calls variants based on the reads aligned to a target, but not necessarily with their precise alignment. It can find SNPs, indels and multi nucleotide polymorphisms (MNPs) in addition to complex events such as composite insertion and substitution events [20]. To check this, we used a myriad of variant calling tools to screen and reach a consensus; for example, Vt was used to identify short variations in the NGS data [21], whereas VarScan, on the other hand, approaches variant detection by aligning the map to multiple locations even as it screens the unique mapped reads for substitutions and indels, thereby detecting multiple reads and converting them into unique SNPs/indels while also determining the total number of reads supporting each allele (reference and variant).

### 2.3. Clustering Coefficient Network Analysis Using cytoHubba

The proteins showing significant changes and interacting between all three different datasets were used to build a gene interaction network using STRING-db, and later were visualized using String [24] and Cytoscape v 3.9.1 [25]. The network was checked for the top-ranking genes built using the expression correlation plugin with a 0.95 correlation as the cut-off value. The cluster IDs for proteins showing similar abundance values as defined in the hierarchical clustering were used to annotate the network to reveal relationships between the different protein groups. Finally, the network analyzer, cytoHubba, was used to define the network measures [26]. A lower contrast (faded yellow/orange) means the rank is lower and a bigger contrast (red/maroon) indicates the rank is greater. We further evaluated the efficacy of network genes for betweenness, closeness, clustering coefficients and stress centrality for the top 10 DEGs. The most common lncRNA and a variant of uncertain significance (VOUS) were checked for the disease phenotypes and the lncRNA was quantitatively measured in our samples by using an RT-PCR for validation.

## 3. Results

Significant DEGs associated with blood disorders were commonly identified across the myocardial, renal and prostate datasets. Between the control and treatment myocardial datasets, apolipoprotein D (APOD ENSG00000189058) was found to be upregulated in the first set (Sc5-LV and HCM515) and downregulated in the other (ND2 and HCM273). It encodes a component of a high-density lipoprotein and shares many similarities with the plasma retinol-binding protein [27]. This glycoprotein has associations with the lipoprotein acyltransferase enzyme lecithin:cholesterol acyltransferase. Breast cysts and androgen insensitivity syndrome (AIS) are two diseases linked to APOD due to cholesterol and sphingolipids being transported/recycled to plasma membranes in the lung (normal and CF), and the transport of glucose and other sugars, bile salts and organic acids; metal ions; and amine compounds are the other associated pathways. CD163 (ENSG00000177575) is downregulated in both myocardial datasets, and is a member of the scavenger receptor cysteine-rich (SRCR) superfamily of scavenger receptors known to be associated with monocytes and macrophages [28]. The gene serves as an acute phase-regulated receptor that helps macrophages remove hemoglobin/haptoglobin complexes and endocytose them, potentially protecting tissues from free hemoglobin-mediated oxidative damage. The protein-encoding gene was shown to act as a bacterial innate immune sensor and a local inflammatory inducer [28]. Furthermore, it is associated with multisystem inflammatory syndrome and histiocytic sarcoma in children. The binding and uptake of ligands by scavenger receptors, hematopoietic stem cells (HSCs) and lineage-specific markers are two mechanisms linked with it. Interestingly, we found a processed pseudogene (ENSG00000274295) that is associated with polymerase (DNA Directed), epsilon 2 and the accessory subunit (POLE2) in these datasets (Figure 2; Appendix A).

Two genes are common between the Sc5-LV/HCM515 and ND2/HCM273 of the myocardial datasets: COL1A1, which codes for the collagen type I α-1 chain (ENSG00000108821), is downregulated in the first dataset and upregulated in the third set; the pro-alpha1 chains of type I collagen, which have two α-1 chains and one α-2 chain, are encoded by this gene. Type I collagen forms fibrils and is found in most connective tissues, including bone, cornea, dermis, and tendon. Osteogenesis imperfecta types I–IV, Ehlers–Danlos syndrome, classical type VIIA, Caffey disease, and idiopathic osteoporosis are all linked to mutations in this gene. Reciprocal translocations between chromosomes 17 and 22 are where genes for platelet-derived growth factor β are located and linked to dermatofibrosarcoma protuberans, a type of skin tumor caused by uncontrolled growth factor expression. Two transcripts have been found as a result of the application of alternative polyadenylation signals. The binding and uptake of ligands by scavenger receptors, as well as VEGFR3 signaling in the lymphatic endothelium, are two linked processes. We also found a significant number of downregulated DEGs in the form of fibroblast growth factor 12 (FGF12; ENSG00000114279), which are associated with the activation of apoptotic and synovial fibroblast pathways regulating a number of biological processes, including embryonic development, cell growth, morphogenesis, tissue repair, tumor growth and invasion, and have extensive mitogenic and cell survival functions. Although it lacks the N-terminal signal sequence seen in the majority of FGF family members, it does include clusters of basic residues that have been shown to behave as a nuclear localization signal. This protein accumulated in the nucleus but was not secreted when transfected into mammalian cells. What remains interesting is that CPNE5 (ENSG00000124772), which encodes a calcium-dependent protein, is downregulated in all three datasets of myocardial function. It harbors an integrin A domain-like sequence in the C-terminus and may regulate molecular events at the interface of the cell membrane and cytoplasm, and is shown to have several alternatively spliced transcript variants encoding isoforms (see Appendix A).

Three genes, AMACR (ENSG00000242110), PCAT14 ((ENSG00000280623) prostate cancer-associated transcript 14), and LTF (lactotransferrin) are frequently compared in studies of prostate data. AMACR (α-methylacyl-CoA racemase) is upregulated in the first two CA prostate datasets only when the latter dataset did not yield any significant DEGs. In addition, various transcript variants with alternative splicing have been identified, such as C1QTNF3 (C1q and tumor necrosis factor-related protein 3), which is known to cause a bile acid synthesis defect; congenital; 4 (Appendix A). As LTF (ENSG00000012223) is downregulated in both sets, it is largely associated with cellular growth and differentiation regulation, cancer formation, and metastasis. It has been recently discovered to have activity against both DNA and RNA viruses, including SARS-CoV-2 and HIV [29]. The two common genes of OLFM4 (ENSG00000102837) and MMP8 (ENSG00000118113) were downregulated in renal WF2-WF4 and upregulated inWF6-WF2. OLFM4 is a novel prognostic predictor as well as therapeutic target for hepatocellular carcinoma [30]. One common lncRNA of OVCH1-AS1 (ENSG00000257599) is upregulated in WF4–WF6 and downregulated in WF6–WF2 (Appendix A).

## 4. Discussion

### 4.1. NONHSAT106693 among Significantly Enriched Genes

One of the favorite candidate DEGs that is often sought is lncRNAs, and what remains compelling is the list of lncRNAs that were upregulated and downregulated across the three datasets. Among them, ENSG00000260604 (lncRNA) is upregulated in both datasets, whereas ENSG00000276980 (intronic complement component 3—C3) and ENSG00000287891 are downregulated. ENSG00000260604 is 1357 nucleotides long (GeneCards, Ensembl, LNCipedia, and Ensembl/GENCODE) and is a well-annotated candidate, with the sense-intronic C3 sequence forming a product of the genes ENSG00000276980.1, ENSG00000276980 and lnc-GPR108-3 [31]. On the other hand, C3 (ENSG00000125730) and LINC02208 (ENSG00000250891) are downregulated in the latter two datasets, as this helps activate the complement system, and the encoded preproprotein is proteolytically processed [32]. Mutations related to this gene are linked to atypical hemolytic uremic syndrome and age-related macular degeneration. The deficiency leads to autosomal recessive and hemolytic uremic syndrome and is widely associated with immune responses, in addition to the lectin-induced complement pathway and peptide ligand-binding receptors. Furthermore, LINC02208 is expressed in tissue samples of the heart [33], even as ENSG00000287891, which is also identified as a novel lncRNA, is downregulated in the latter two myocardial datasets.

Interestingly, a novel lncRNA (ENSG00000285534) is downregulated in both sets of R2-R2_pt and R4-R4_pt, whereas CXCL8 (C-X-C motif chemokine ligand 8 (ENSG00000169429)), which is associated with melanoma and bronchiolitis is downregulated in R5-R5_pt and upregulated in R6-R6_pt. They aid in immune response CCR3 signaling in eosinophils and cytokine signaling, and produce a protein that belongs to the CXC chemokine family (encoded by IL-8), a key mediator of the inflammatory response. Mononuclear macrophages, neutrophils, eosinophils, T lymphocytes, epithelial cells and fibroblasts all produce IL-8, which acts as a chemotactic factor, directing neutrophils to the infection site. In addition to participating in the proinflammatory signaling cascade with other cytokines, it may be likely that the overproduction of such proinflammatory proteins are assumed to be the source of the cystic fibrosis-related lung inflammation which may contribute to coronary artery disease and endothelial dysfunction. Tumor cells release this protein, which promotes tumor motility, invasion, angiogenesis and metastasis. This chemokine also has angiogenic properties. Higher levels of IL-8 are positively connected with the increased severity of numerous illness outcomes, and IL-8 binding to one of its receptors (IL-8RB/CXCR2) enhances blood vessel permeability (e.g., sepsis). On the other hand, LEF1 (ENSG00000138795) is upregulated in both datasets (R5 and R6) and three other genes, *viz*., G0S2 (ENSG00000123689), HSD11B1 (ENSG00000227591) and PTGS2 (ENSG00000073756), were commonly downregulated in the R4 and R5 sets. G0S2 (G0/G1 Switch 2) is a protein-coding gene located in the mitochondria involved in the extrinsic apoptotic signaling pathway, which plays a role in the positive regulation of the extrinsic apoptotic signaling pathway regulating Van der Woude syndrome. HSD11B1 (ENSG00000227591), or more specifically, HSD11B1-AS1 (HSD11B1 antisense RNA 1), is a lncRNA and is associated with the cortisone reductase deficiency. Besides this, the genes POSTN (ENSG00000133110); EPDR1 (ENSG00000086289); and SFRP4 (ENSG00000106483) were upregulated in the third set of myocardial data and upregulated in the second set of prostate data. Periostin is a secreted protein that induces cell attachment and spreading; plays a role in cell adhesion; and its differential expression is known to regulate T2-high asthma, myocardial-infarction-regulating heparin binding and cell-adhesion molecule binding [34]. LTF (ENSG00000012223) is downregulated in the second set of prostate and treated renal data (WF2-WF4), with a novel lncRNA NONHSAT106693 (ENSG00000287903) shown to be upregulated in the third set of myocardial data and upregulated in the treated renal data.

### 4.2. Heatmaps Showed Distinct Gene Expression Profiles with a Variance in Principal Components

When dealing with multivariate data, using a matrix in high-throughput experiments for ascertaining gene expression patterns is of considerable interest. From the varied datasets, we have attempted to identify the transcriptome expression by comparing the control and treated datasets to check their variance (Appendix A). Overall, we obtained 392 DEGs, of which 172 genes were upregulated and 220 genes were downregulated (Appendix A). In myocardial datasets, 10 genes were upregulated when comparing the first datasets (Sc5-LV and HCM515), whereas 6 genes were upregulated and 28 genes were downregulated in the second datasets and 28 genes were upregulated and 24 genes were downregulated in the third datasets (N102_LV and HCM506). Among the prostate data, 14 genes were upregulated in the first datasets (10N and 10T), 34 genes were upregulated in the second datasets (2N and 2T) and there were no significant DEGs found in the third datasets (3N and 3T). In comparing renal datasets, the test group and post-therapy group were analyzed, of which 9 genes were upregulated and 31 genes were downregulated in the first datasets (R2 and R2_pt); 7 genes were upregulated and 30 genes were downregulated in the second datasets (R4 and R4_pt); 13 genes were upregulated and 21 were downregulated in the third datasets (R5 and R5_pt); and 19 genes were upregulated and no genes were downregulated in the fourth datasets (R6 and R6_pt). The results of the control group include 9 genes that were upregulated in the first datasets (WF2 and WF4); 4 genes that were upregulated and 11 genes that were downregulated in the second datasets (WF4 and WF6); and 19 genes that were upregulated and 1 gene that was downregulated in the third datasets (WF6 and WF2).

### 4.3. CytoHubba Yielded Top Niche Ranks with Variant Analysis Showing No Mutations Attributed to DEGs

We sought to ask whether the common DEGs form top niche ranks in all three datasets. To check this, we imported the network of DEGs (Figure 3A) into Cytoscape and visualized all other networks, such as closeness, betweenness, stress and clustering genes, in cytoHubba. The network with the top 10 genes yielded a rank list indicating the top niche genes associated through the clustering coefficient. Although the color indicates the score of the genes interacting in the network, we found that among the top-ranking genes, four DEGs are known to be associated with VK deficiency (Figure 3B). COL1A2 (α-2 type I collagen) is one of the profibrotic genes that expresses osteocalcin in the liver [35] and is also used as a marker of cardiac fibrosis [36]. POSTN (periostin) is one of the VK-dependent proteins, which is majorly involved in hematopoiesis [37], myocardial infarction, fibrosis and bone health [38]. SFRP4 (secreted frizzled-related protein 4) is involved in bone mineral density, which is related to osteoporosis [39]. The EPDR1 (ependymin-related 1) gene produces a type II transmembrane protein that is related to the protocadherins and ependymoma, two families of cell-adhesion molecules. This protein may have a role in calcium-dependent cell adhesion, according to gene expression studies in brain tissue [40]. LTF (lactotransferrin) has been found to have an effect on host immunological responses and has a potential antagonistic pleiotropy, suggesting that it may be protective against caries in addition to being predisposed to localized aggressive periodontitis [41]. The above comorbidities are related, and hence, these four genes are considered the hub genes associated with VK deficiency. With this proposition, we determined the extent of lncRNAs in the network, and, as a result, we could plot the lncRNA top-ranking DEGs as well (Figure 3C; Appendix A). Taken together, we found NONHSAT106693 to be a novel lncRNA with a large expression in the testis, indicating that it could be associated with all three, viz., VK, renal and PCa (FPKM/TPM: 0.14; Figure 3D). We have reconfirmed the expression in vivid samples of VK in-house (data shown).

### 4.4. Variants of Unknown Significance

The variants called from these DEGs were further compared with five different tools: VarScan, SAMtools, FreeBayes, Vt and GATK. To check whether any DEGs harbor the pathogenic variants, we compared ClinVar pathogenic variants of VK with a list of a significant number of variants (Appendix A). Among them, 37 variants were found to have a match with the ClinVar data, which were associated with CFTR, ESR1, GGCX, ATP8B1, VWF, GLA and F8. Although CFTR (chr 7) has been afflicted in both myocardial (rs397508397) and prostate control (rs75789129) samples from SAMtools and Vt, ESR1 was shown to be seen in both the renal control and treated samples, and the F8 was seen only in the myocardial treated samples. On the other hand, rs563109158 (T > C) was found in the gene GGCX, which is an extremely rare variant of uncertain significance; it was found to be common in all the three sets, implying that this is predisposed in an ostensibly large population (C = 0.000318/6 (ALFA)/C = 0./0 (TWINSUK)/C = 0.000223/1 (Estonian)/C = 0.000404/107 (TOPMED)/C = 0.000407/57 (GnomAD)/C = 0.000519/2 (ALSPAC)/C = 0.001002/1 (GoNL)). This was further considered as a candidate for the classification of disease prevalence and penetrance estimates and was, therefore, classified as a variant of unknown significance (VOUS). Taken together, none of the DEGs seem to be commonly enriched from our prostate RNA-Seq datasets screened in-house (data unpublished). This VOUS was common, particularly in the control sets (Sc5-LV; ND2) and the myocardial treated set (HCM506); the control set of 2T and the treated set of 3T in prostate samples; the test group of R4, R5 and R6 and the post-therapy group of R4_pt and R5_pt; and the control group of WF2 in Renal datasets. This indicates that it is found across different phenotypes. The interpretation was reported as the uncertain significance and the variant condition was identified as VK-dependent clotting factors combined with the deficiency of type 1 with no citations found in ClinVar, and therefore, we have validated the finding using Sanger sequencing from our PCa/VK cohort ). The other genes, ATP8B1, GGCX, GLA and VWF, were identified in all three control and treated samples as the identified results were taken for further analysis of molecular docking and simulation studies.

## 5. Conclusions

Vitamin K (VK) plays an important role in human metabolism. In this work, we investigated whether any common DEGs were significantly enriched among various datasets and, if so, whether or not the variants in them are noteworthy for VK disease phenotypes, viz., myocardial, renal and prostate cancer. Although we found a large number of lncRNAs among the DEGs, NONHSAT106693 was found to be a significantly enriched lncRNA across the renal and myocardial datasets, implying that it plays an important role in atypical hemolytic uremic syndrome and age-related macular degeneration. Our work also emphasizes the role of variants of unknown significance (VOUS) in these phenotypes, especially the common variant, viz., rs563109158, seen in GGCX, which is associated with VK-dependent clotting factors. There is room for analyzing more datasets associated with VK, coagulation and blood disorders, which would set a precedent in screening pathogenic and, perhaps, unique variants/VOUSs as the downstream analysis and development of NGS panels for rare blood disorders and VK deficiencies are on the rise.

## Figures and Tables

**Figure 1 genes-13-02078-f001:**
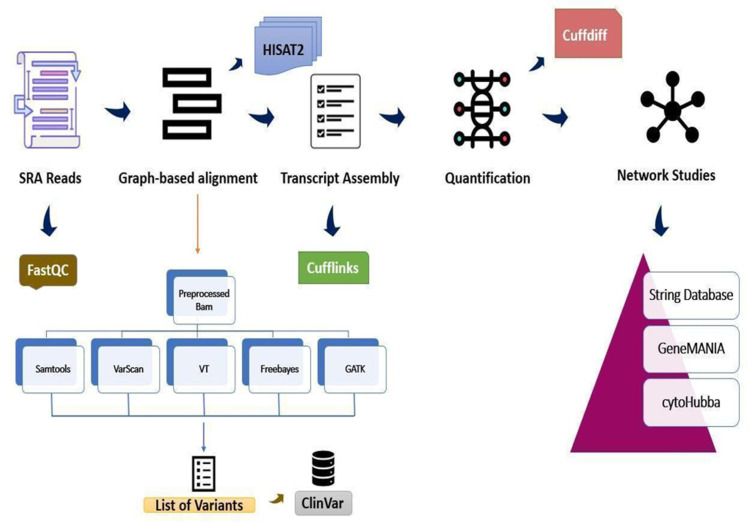
Workflow of the RNA-Seq pipeline. The pipeline for differential gene expression analysis of RNA-Seq data. In the variant calling step, five different approaches, *viz*. SAMtools, VarScan, Vt, FreeBayes and GATK, were utilized to identify significant SNPs.

**Figure 2 genes-13-02078-f002:**
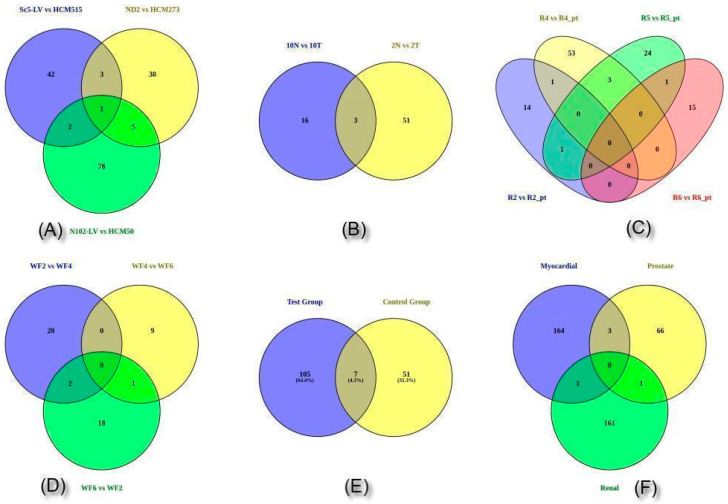
Differentially expressed genes between control and treated in all three datasets, viz., (**A**) (control and treated) myocardial, (**B**) (control and treated) prostate, and (**C**) test (treated) group of renal data; (**D**) control group of renal data, (**E**) control and test group of renal data, and (**F**) common in all three, myocardial, renal and prostate data.

**Figure 3 genes-13-02078-f003:**
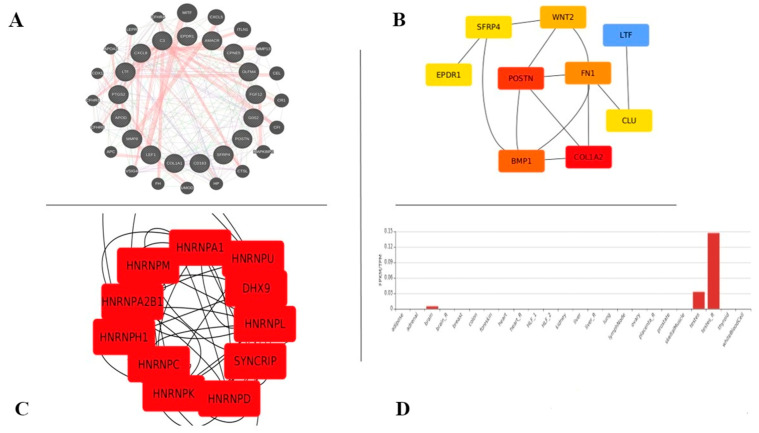
(**A**) The network illustrates the interaction between the differentially expressed genes in all three datasets (myocardial, prostate and renal), and it was constructed with GeneMANIA. (**B**) The network image represents the clustering coefficient of common DE genes from all three datasets. The image was generated with cytoHubba. (**C**) The top ranks of lncRNAs in clustering coefficient networks. (**D**) The graph shows the tissue expression of common novel lncRNA, and it is generated with the NONCODE database.

## Data Availability

The PCa datasets are deposited at the Sequence Read Archive (SRA) with project ID: PRJNA616165, and the renal datasets with ID: PRJNA854340.

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
