# Peer review of "Inferring bona fide Differentially Expressed Genes and Their Variants Associated with Vitamin K Deficiency Using a Systems Genetics Approach"

_genes, 2022, doi:10.3390/genes13112078_

Round 1
Reviewer 1 Report
In this manuscript, Rajagopal S. et.al. investigate differencial expression of genes and variants in various myocardial, prostate and renal diseased subpopulations with vitamin k deficiency. While the study aims to address the analysis in a rigorous way, there are few points that needs to be addressed.
1. For alignment STAR 2 pass is a better choice to tool over hisat2 for followup variant calling. Since this paper has a high focus on that, they should at least perform this in a sample dataset to make sure they not getting a different result.
2. "we used a myriad of variant calling tools to screen and reach consensus,” How to reach consensus? Common between them? A particular scoring system?
3. Figure 2: a better clarifications on the groups are needed. How are they chosen? what do they represent?
4. No potential mechanism pointed out in the conclusion. the discussion and conclusion needs marked improvement. At this point it looks more like a data dump than some insights.
Author Response
- For alignment STAR 2 pass is a better choice to tool over hisat2 for followup variant calling. Since this paper has a high focus on that, they should at least perform this in a sample dataset to make sure they not getting a different result.
Thank you very much for the constructive comments. This paper focuses on both differential expressed genes and SNPs. We have chosen HISAT2 because we don’t want to miss the splice junctions in the genome, which is an important modulator of gene expression. From our previous benchmarking done with these datasets, hisat2 fares well.
- "we used a myriad of variant calling tools to screen and reach consensus,” How to reach consensus? Common between them? A particular scoring system?
Thank you for the comments. We used five different tools including samtools, VT, freebayes, VarScan, and GATK, to identify the consensus SNPs. All SNPs were compared to the ClinVar Databases’ list of Vitamin K Genes. All five tools have found the consensus SNP. The consensus SNP have been identified using all the five aforementioned tools.
- Figure 2: a better clarifications on the groups are needed. How are they chosen? what do they represent?
Thank you for the comments. The Figure.2 represents the inter and intra group of datasets and we have now explained elaborately.
- No potential mechanism pointed out in the conclusion. the discussion and conclusion needs marked improvement. At this point it looks more like a data dump than some insights.
Thank you. We have paraphrased and changed it. Kindly see.
Reviewer 2 Report
This article fits the special issue “discoveries in sequencing data” very well. Rajagopal and colleagues used a systems genetics approach to analyze several RNA sequencing databases and identified a couple of differentially expressed genes that may relate to vitamin K deficiency. It is a good approach to identifying novel biomarkers. But the relationship between differentially expressed genes and vitamin K deficiency needs to be further explained, as the purpose of this paper is to study vitamin K deficiency-related genes.
Comments are as follows:
(1) Although vitamin K deficiency was demonstrated to be associated with several diseases, it won’t be a direct cause of myocardial infractions, renal failure, and prostate cancer. It would be better to explain more about why chose the databases of these diseases, or any other common points of this disease.
(2) It would be better if you can provide a table that includes all differentially expressed genes found in this project. List all the expression pattern (upregulated or downregulated in which disease) and their relationship with vitamin K deficiency. Vitamin K deficiency is always caused by genes that will affect the quantity or utilization of vitamin K, such as GGCX and VKOR. The differential expression of the individual substrate (vitamin K-dependent proteins) is only associated with the function itself, but not vitamin K deficiency.
(3) As you mentioned in the last sentence in 3.3, qRT-PCR of lncRNA has been done by the result is now shown. This will be direct and the only evidence to support that lncRNA is related to vitamin K deficiency. This result is necessary for this paper.
(4) I didn’t see Supplementary Table 2 and Supplementary Figure 3, which were mentioned in the discussion part.
Author Response
(1) Although vitamin K deficiency was demonstrated to be associated with several diseases, it won’t be a direct cause of myocardial infractions, renal failure, and prostate cancer. It would be better to explain more about why chose the databases of these diseases, or any other common points of this disease.
Thank you for the comments. As we are mainly focusing on the lncRNA, we deem to find well known characterized datasets have chosen the myocardial data sets which is the work related to long non-coding RNA in human hypertrophic cardiomyopathy, prostate and renal data. We added the following:
There are, indeed, associated VK phenotypes such as thrombosis, thrombocytopenia, myocardial infarction, renal failure, prostate cancer which are used to ascertain the differential expressed gene (DEG) profiles
(2) It would be better if you can provide a table that includes all differentially expressed genes found in this project. List all the expression pattern (upregulated or downregulated in which disease) and their relationship with vitamin K deficiency. Vitamin K deficiency is always caused by genes that will affect the quantity or utilization of vitamin K, such as GGCX and VKOR. The differential expression of the individual substrate (vitamin K-dependent proteins) is only associated with the function itself, but not vitamin K deficiency.
Thank you for the comments. Apologies, if those were missing last time. We have listed all the differential expressed genes in the supplementary tables. The vitamin K‑dependent proteins may exert their functions following γ‑carboxylation with vitamin K, and different vitamin K‑dependent proteins may exhibit synergistic effects or antagonistic effects on each other.
(3) As you mentioned in the last sentence in 3.3, qRT-PCR of lncRNA has been done by the result is now shown. This will be direct and the only evidence to support that lncRNA is related to vitamin K deficiency. This result is necessary for this paper.
Thank you for the comments. We have updated the results and added a supplementary information.
(4) I didn’t see Supplementary Table 2 and Supplementary Figure 3, which were mentioned in the discussion part.
Thank you for the comments. We have uploaded the supplementary Table 2 and supplementary Figure.3 separately